# Perceived Social Support Mediates the Relationship between Use of Greenspace and Geriatric Depression: A Cross-Sectional Study in a Sample of South-Italian Older Adults

**DOI:** 10.3390/ijerph20085540

**Published:** 2023-04-17

**Authors:** Elisabetta Ricciardi, Giuseppina Spano, Luigi Tinella, Antonella Lopez, Carmine Clemente, Andrea Bosco, Alessandro Oronzo Caffò

**Affiliations:** 1Department of Educational Sciences, Psychology, Communication, University of Studies of Bari, 70122 Bari, Italy; elisabetta.ricciardi@uniba.it (E.R.); giuseppina.spano@uniba.it (G.S.); luigi.tinella@uniba.it (L.T.); a.lopez@unifortunato.eu (A.L.); carmine.clemente@uniba.it (C.C.); andrea.bosco@uniba.it (A.B.); 2Faculty of Law, Giustino Fortunato University, 82100 Benevento, Italy

**Keywords:** greenspace, depression, perceived social support, ageing, structural equation modeling

## Abstract

A growing body of evidence is suggestive for the beneficial role of contact with greenspace (e.g., use of greenspace, visual access to greenspace, etc.) on mental health (e.g., depression, anxiety, etc.). In addition, several studies have pointed out the benefits of social support and social interaction on psychological wellbeing. Even if evidence on the association between contact with greenspace and perceived social support were mixed, it was supposed that the use of greenspace could enhance social interactions and perceived social support, especially among older adults. The present study aims to explore the effect of use of greenspace on geriatric depression in a sample of South-Italian older adults and the mediating role of perceived social support in this association. A structural equation model was tested in a sample of 454 older adults (60–90 years old) residing in the Metropolitan Area of Bari, Apulia. The fit indices revealed the goodness of fit of the model (CFI = 0.934; TLI = 0.900; IFI = 0.911; NFI = 0.935; RMSEA = 0.074; SRMR = 0.056). Results showed that the use of greenspace was inversely associated with geriatric depression through perceived social support. These findings underlined the relevance of perceived social support on the pathway linking use of greenspace and geriatric depressive symptoms. This evidence may be useful to policymakers to plan interventions for promoting physical access to greenspace and social participation in an age-friendly city framework.

## 1. Introduction

Worldwide, the estimate of prevalence of depression in older people was assessed to be around 28.4%, and in Europe was to be around 21.1% [1]. Depression is a common mental health condition characterized by depressed mood, such as sadness feelings and hopelessness, and loss of interest or pleasure in everyday activities. Patterns of insomnia or hypersomnia, poor concentration, weight loss or gain, worthless feelings, and thoughts of death may be experienced during this condition [2]. During ageing, people become more vulnerable to develop depressive symptoms, due to increasing in difficulties in physical abilities, chronic health problems, loneliness, and social disconnection [3,4,5,6].

As is well-established, social and environmental features may positively affect older people in a protective way against age-related physical and mental impairments [6]. For instance, contact with urban greenspace (e.g., urban parks, community gardens, etc.) is recognized to provide health benefit across lifespan [7,8,9,10,11,12,13,14]. The positive association between contact with greenspace and several health outcomes during ageing was supported by a growing number of studies [15,16,17,18]. Despite that, few evidence is available on the association between contact with urban greenspace and mental health during ageing [19,20,21,22]. Specifically, few studies on the association between contact with urban greenspace and reduction in depression symptoms were available on older adults [23,24,25]. For instance, Banay et al. [23] found an inverse association between residential greenspace and depression symptoms in a sample of older women. Similar results were found by Perrino et al. [24] in a sample of older adults. Instead, Pun et al. [25] found no association between neighborhood greenness and depressive scores. A recent meta-analysis [26] investigated the association between short-term exposure to greenspace and depression symptoms. Among the reviewed studies (N = 33), only one included older adults. Furthermore, a small effect was found for the association between short-term exposure to greenspace and depressive mood.

Overall, Markevychh et al. [27] summarized in a unifying conceptual framework several mechanisms underlying mental health benefits associated with greenspace. One of the most important concerns the restoring capacities of greenspace, i.e., the so-called “restorativeness”, based on the Attention Restoration Theory (ART) and on the Stress Reduction Theory (SRT) [28,29]. The ART [29] provides a theoretical framework and related predictions for the cognitive impact of urban greenspace. Similarly, the SRT [28] provides theory and predictions for the affective impact of urban greenspace. Specifically, the SRT assume that contact with urban greenspace through the activation of parasympathetic system reduces stress, autonomic arousal, and enhances positive emotions, since people have evolved with an innate preference for green environments [28,30]. Other underlying variables are physical exercise and social benefits, including social connectedness, social interactions, and social support [27,31,32,33]. Especially, perceived social support was found to be involved in the relationship between contact with urban greenspace and mental health benefits [34,35,36]. Kwon et al. [36] found that social support was a mediator in the relationship between urban greenspace and happiness. Similar results were found by Gascon et al. [34]. Nevertheless, a recent systematic review showed that mixed or non-significant evidence was found for social support as intervening variable in the aforementioned association.

Additionally, human engagement with greenspace includes indirect contact with greenspace, such as viewing greenspace from a distant point, and direct contact with urban greenspace (i.e., use of greenspace), such as visiting and spending time in greenspace [37]. Nevertheless, evidence on the effect of visiting greenspace on mental health were still limited [38].

In light of the above-mentioned lack in previous research a cross-sectional study was carried out. The present study attempts to extend knowledge on the relationship between self-reported direct contact with greenspace (i.e., use of greenspace) and self-reported depression symptoms, and to explore one of the potential underlying mechanisms of this association (i.e., perceived social support). The aims of this study were: (a) to investigate the relationship between use of greenspace and depression symptoms during ageing, and (b) to investigate the role of perceived social support in the aforementioned relationship.

## 2. Materials and Methods

### 2.1. Participants and Procedure

A convenience sample of older adults residing in the metropolitan area of Bari (Apulia) were recruited between January 2022 and April 2022, with the support of a proxy informant. Proxy informants were recruited among undergraduate students, postgraduate students, and trainees. They contacted and invited older adults to be enrolled in the study. Inclusion criteria to the enrollment in the study were: (a) aged 60 or older, (b) living in the metropolitan area of Bari, (c) no history of neurological or psychiatric disease, and (d) no visual or hearing loss. The total sample included 454 older adults (ranging age between 60 to 90) (mean = 73.00, sd = 7.43). An informed consent form was obtained by all the participants. The study was approved by the local ethical committee, and it was carried out in accordance with the Helsinki declaration and its later amendments. Each participant was administered, by a well-trained research assistant, an *ad hoc* protocol including socio-demographic questionnaire, self-reported questionnaires, and single-item questions.

### 2.2. Measures

#### 2.2.1. Use of Greenspace

To assess frequency of visits in greenspace, a single item asking participants how often they usually visit urban greenspace (i.e., “how often do you visit greenspace in your neighborhood, such as parks, community gardens?”) was used. The item was rated on a 5-point Likert scale ranging between 1 (almost never) and 5 (several times a week). To assess time spent in greenspace an additional single item asking participants to indicate time in minutes spent in urban greenspace (i.e., “how long do you stay there when you visit it?”) was used.

#### 2.2.2. Perceived Social Support

To assess perceived social support, the Duke-UNC Functional Social Support Questionnaire (FSSQ) [39] was employed. The items included in the original English validation were translated in Italian using the forward back translation method with the support of translators with qualifications. Eight 5-point Likert scale (1, much less I would like; 5, as much I would like) items compose the questionnaire. In these data, the Cronbach’s α and the McDonald’s ω of FSSQ were, respectively, 0.84 and 0.85. Scale (Appendix A) and item reliability statistics (Appendix A) were reported in Appendix A. The FSSQ is made of two dimensions, which correspond to two subscales. Three items refer to affective support subscale (e.g., “I have people who care happens to me”), and five items refer to confidant support subscale (e.g., “I get chances to talk to someone about I trust about my personal or family problems). Higher values indicate a higher level of perceived social support.

#### 2.2.3. Geriatric Depression

The Geriatric Depression Scale −15 item (GDS) to assess geriatric depression was used. GDS is a self-reported questionnaire, validated both in English and Italian [40,41,42]. This scale is composed of 15 items with a dichotomous answer (i.e., yes/no), and 3 dimensions corresponding to 3 subscales [42]. In these data, the Cronbach’s α and the McDonald’s ω of FSSQ were, respectively, 0.73 and 0.74. Seven items refer to geriatric depression affect subscale (e.g., “Often feel hopeless”), four items refer to life satisfaction subscale (e.g., “Satisfied with life”), and four items refer to withdrawal subscale (e.g., “Prefer to stay at home”). Higher scores indicate higher levels of symptoms of depression.

### 2.3. Statistical Analyses

Data was analyzed using R software [43]. Descriptive statistics were performed on sociodemographic characteristics of participants. Pearson correlation coefficients were calculated to identify general patterns of our variables (i.e., frequency of visits in greenspace, time spent in greenspace, geriatric depression, and perceived social support) analyzing the total score for each of them. Figure 1 indicates the graphical representation of the theoretical model proposed. A structural equation modeling (SEM) was employed to explore our hypotheses on structural paths between latent variables, using lavaan package [44] on R statistical software [43]. Geriatric depression symptoms (i.e., latent variable) as outcome, use of greenspace (i.e., latent variable) as predictor, and perceived social support as mediator were tested in the model, adjusting for age and education. Goodness of fit was evaluated considering the following indices and related cut-offs: the Comparative Fit Index (CFI) as provided by Hu and Bentler [45], the Tucker–Lewis Index (TLI), the Incremental Fit Index (IFI) and the Normed Fit Index (NFI) indicative of a good fit if ≥0.90 [46,47], the Standardized Root Mean Square Residual (SRMR) indicative of an acceptable fit if <0.08, and the Root Mean Square Error of Approximation (RMSEA) indicative of a good fit if <0.08 [48], and the Chi-squared value (χ^2^), divided by the degree of freedom (**χ**^2^/df), indicative of a good fit if less than 5 [49].

## 3. Results

### 3.1. Descriptive Statistics

In Table 1 means, standard deviations, and correlation coefficients (i.e., Pearson coefficients) of the variables considered are shown. Significant correlations ranged between 0.940 (*p* < 0.001) and −0.441 (*p* < 0.001). Results indicated significant positive correlation between frequency in GS and time spent in GS (r = 0.573; *p* < 0.001), between subscale “affect” of GDS and subscale “life satisfaction” of GDS (r = 0.460; *p* < 0.001), between subscale “affect” of GDS and subscale “withdrawal” of GDS (r = 0.387; *p* < 0.001), between subscale “life satisfaction” of GDS and subscale “withdrawal” of GDS (r = 0.329; *p* < 0.001), and between subscale “affective support” of FSSQ and subscale “confidant support” of FSSQ (r = 0.527; *p* < 0.001).

### 3.2. Model Testing

Figure 2 indicates the graphical representation of the model tested.

Regarding the goodness of fit of the proposed model, the CFI was 0.934, the TLI was 0.900, the IFI was 0.935, and the NFI was 0.911. All these values were indicative of a barely acceptable fit of the model. The SRMR was 0.056 and the RMSEA was 0.074 (90% CI: 0.052–0.084), and they indicated a good fit of the model. The ratio between chi-square value and its degrees of freedom (3.50) also indicated a good model fit. Overall, the model can be considered acceptable. In addition, all factor loadings on the predictor (i.e., use of greenspace), on the mediator (i.e., perceived social support), and on the outcome (i.e., geriatric depression) were statistically significant (at least with *p* < 0.01).

Completely standardized solution coefficients of structural paths were presented in Table 2.

Use of greenspace was positively associated with perceived social support (β = 0.181; se = 0.061; *p* = 0.003), i.e., the more older adults used urban greenspace, the more they perceived to be socially supported. Use of greenspace was found to be not significantly associated with geriatric depression, i.e., there were no differences in geriatric depressive symptoms between older adults who had higher rates and older adults who had lower rates of use of greenspace. Perceived social support was found to be negatively associated with geriatric depression (β = −0.433; se = 0.059; *p* < 0.001), i.e., the more older adults perceived to be socially supported, the less they reported geriatric depression symptoms. Both education (β = −0.263; se = 0.059; *p* < 0.001) and age (β = 0.226; se = 0.054; *p* < 0.001) were associated with geriatric depression.

Completely standardized solution coefficients of direct, indirect, and total effect were presented in Table 2, as well. The direct effect on the path between use of greenspace and geriatric depression was found not to be significant. The indirect effect on the path between use of greenspace and geriatric depression through perceived social support was significant (β = −0.078; se = 0.029; *p* = 0.007), as well as the total effect of the whole model (β = −0.142; se = 0.060; *p* = 0.019).

## 4. Discussion

The present study attempted to provide more knowledge on the relationship between contact with urban greenspace and depressive symptoms during ageing. The present paper aimed to investigate this relationship using measures of direct contact with urban greenspace (i.e., use of greenspace). The use of greenspace was assessed considering frequency of visits in greenspace and time spent in greenspace weekly by participants involved in the study. Furthermore, this study aimed to investigate the role of perceived social support in the above-mentioned association.

Overall, adopting a structural equation modeling statistical framework, it was found that direct contact with greenspace positively affects mental health in older adults, but only through perceived social support increased by greenspace. In our study, older people who had higher rates of use of greenspace than others, experienced more perceived social support, and older people who had higher levels of perceived social support, experience fewer geriatric depression symptoms. Previous studies revealed that visiting and spending time in greenspace provided improvements in perceived mental health [50,51,52]. For instance, van den Berg et al. found a positive association between spending time in greenspace and mental health and vitality. This study confirmed that the aforementioned association was independent of cultural and climatic contexts. Furthermore, White et al. [52] found that the likelihood of reporting wellbeing was greater in people who spent at least 120 min per week in greenspace than in those who reported less time spent, even if positive association peaked for people who spent 300 min per week in greenspace. As opposed to, in the present study it was observed that use of greenspace was not directly associated with geriatric depression symptoms. Direct contact with urban greenspace may not be a requisite for improving health.

Indeed, it was observed that there was an indirect association between use of greenspace and geriatric depression. Our findings confirm the general hypothesis that contact with urban greenspace provided mental health benefits through several underlying mechanisms. This is in line with previous results on indirect association between use of urban greenspace and mental health [53], which indicated that visiting urban greenspace was associated with greater human wellbeing through several intervening variable (e.g., ego-depletion). Specifically, it was observed the mediating role of perceived social support in the association between contact with urban greenspace and geriatric depression as supported by previous research [35,54]. For instance, Dadvand et al. [54] found that perceived social support mediated the association between greenspace exposure and general health. Social support and perceived social support were found as mediators in the relationship between greenspace and mental health in several studies as confirmed by a recent review [35]. One possible explanation for the underlying mechanism of perceived social support regards the well-established social benefits (i.e., social interactions, social cohesion, etc.) of contact with urban greenspace. One may suppose that social interaction opportunities offered by the use of greenspace performs a crucial role in promoting neighborhood social cohesion [55], thus enhancing perceived social support of older people. Our findings confirmed the protective role of perceived social support on geriatric depression symptoms [56,57]. Lastly, mental health benefits provided by contact with urban greenspace were widely proved [21,58,59].

To our knowledge, limited evidence on older adults was available. Our study highlighted the relevance of contact with urban greenspace for older people [24]. Furthermore, the beneficial role of contact with urban greenspace on mental health for older adults was confirmed, as suggested by previous studies [34,60].

This study had some limitations. First, our data were cross-sectional, thus preventing the causal relationship between use of greenspace and geriatric depression. Moreover, since study results refer to a specific population study (i.e., older adults from the South of Italy) and to a relatively small sample size, these findings may suffer from a lack in generalizability to other population targets (e.g., young and mature adults). Furthermore, self-reported measures were used for assessing both the use of greenspace and geriatric depression symptoms and perceived social support, as well. Self-report measures may be affected by an underestimation or overestimation bias with respect to internal disposition, and recall bias with respect to previous behaviors. Lastly, in the present study, greenspace attributes (e.g., pleasantness, aesthetic quality, layout, etc.), individual characteristics (e.g., connectedness to nature, urban-nature orientedness, etc.), which may be predictors of propensity to visit urban greenspace and of mental health benefits of greenspace were left out [61,62,63,64]. For instance, as suggested by previous studies [61,64], visit frequency in urban greenspace and mental health benefit of greenspace may be influenced by park features, such as size, layout, pleasantness, and items involved in the park area. Furthermore, Ojala et al. [63] and Sella et al. [65] suggested that individual characteristics (e.g., urban-nature orientedness) may affect restorativeness provided by contact with greenspace. These issues should be addressed in future studies. This study may be replicated by: (a) exploring the proposed structural relationship in different population targets (e.g., young and mature adults), (b) using a Global Positioning System (GPS) device to objectively measure the time that participants spend in greenspace, (c) using objective data for assessing geriatric depression symptoms (e.g., medical prescriptions, psychiatric diagnosis) and for social support (e.g., sociometric tools), and (d) considering some individual characteristics, such as measures of spatial cognition based on familiar [66] and unfamiliar locations, and (e) greenspace attributes.

## 5. Conclusions

The present study attempts to extend knowledge on mental health benefits of contact with urban greenspace during ageing. These findings highlighted the beneficial role of greenspace use on geriatric depression, through the mediation of perceived social support. These results confirmed that visiting and spending time in greenspace enhances perceived social support among older adults, which in turn has a positive effect on geriatric depression. Future research may take into account differences in levels of urbanization and in rates of urban greenspace in other countries to confirm and extend the present results. Lastly, further studies could investigate the role of urban greenspace characteristics (i.e., quality of greenspace, biodiversity, etc.), proximity to greenspace, ecosystem services and disservices, which are closely linked to direct contact with urban greenspace. Furthermore, these results may be of interest to policy makers in light of the relevance of geriatric depression for the public expenditure. Indeed, geriatric depression is recognized as a significant public health issue which is strictly associated with economic burden for society. Adequate strategies for prevention of geriatric depression are needed. For instance, in order promote mental health and healthy lifestyle during the ageing, age-friendly environments and tailored nature-based solutions could be a relevant target to be pursued by policy makers and offered to the target population. Moreover, social policies could consider the beneficial role of greenspace on mental health and these findings could be addressed in future urban planning.

## Figures and Tables

**Figure 1 ijerph-20-05540-f001:**
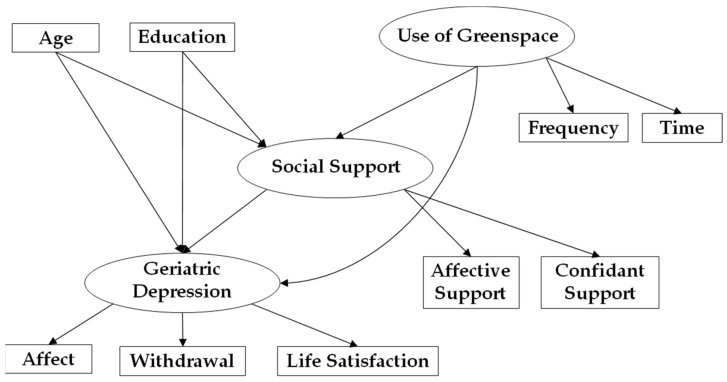
Proposed theoretical model.

**Figure 2 ijerph-20-05540-f002:**
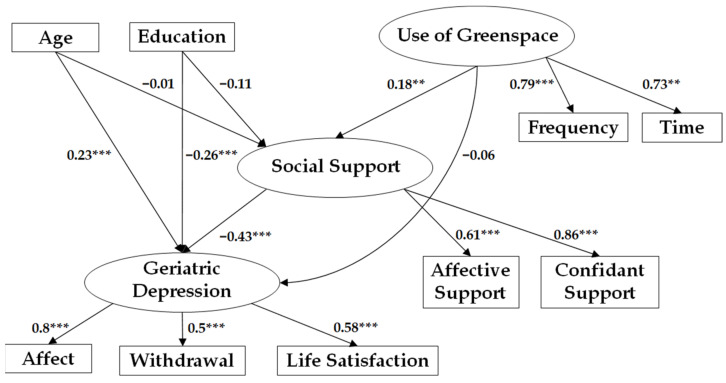
Path diagram of the Structural Equation Model tested. ** *p* < 0.01; *** *p* < 0.001.

**Table 1 ijerph-20-05540-t001:** Means, Standard Deviations, and Correlations of variables included in the study.

Variables	Mean	Standard Deviation	Age	Education	Frequency in GS	Time Spent in GS	GDS Affect	GDS Life Satisfaction	GDS Withdrawal	GDS Total	FSSQ Affective Support	FSSQ Confidant Support	FSSQ Total
Age	73.00	7.43	—										
Education	9.09	5.06	−0.441 ***	—									
Frequency in GS	2.30	1.62	−0.106 *	0.131 **	—								
Time spent in GS	31.2	41.5	−0.077	0.029	0.573 ***	—							
GDS Affect	1.79	1.61	0.303 ***	−0.310 ***	−0.101 *	−0.072	—						
GDS Life Satisfaction	1.06	1.21	0.090	−0.092	−0.144 **	−0.093 *	0.460 ***	—					
GDS Withdrawal	1.28	0.95	0.201 ***	−0.136 **	−0.147 **	−0.090	0.387 ***	0.329 ***	—				
GDS Total	4.32	3.05	0.265 ***	−0.245 ***	−0.154 ***	−0.101 *	0.853 ***	0.757 ***	0.670 ***	—			
FSSQ Affective Support	12.5	2.72	0.106 *	−0.116 *	0.034	−0.020	−0.164 ***	−0.256 ***	−0.008	−0.192 ***	—		
FSSQ Confidant Support	18.2	5.12	−0.013	−0.057	0.121 **	0.148 **	−0.265 ***	−0.282 ***	−0.176 ***	−0.322 ***	0.527 ***	—	
FSSQ Total	30.7	6.97	0.032	−0.088	0.101 *	0.101 *	−0.259 ***	−0.306 ***	−0.136 **	−0.313 ***	0.781 ***	0.940 ***	—

Note: GDS = Geriatric Depression Scale; GS = Greenspace; FSSQ = Duke UNC functional social support questionnaire. Note. * *p* < 0.05, ** *p* < 0.01, *** *p* < 0.001.

**Table 2 ijerph-20-05540-t002:** Standardized effects, standard errors, z scores, and *p* values for each outcome variable, direct, indirect, and total effect of use of GS on geriatric depression.

Effect	Standard Estimate	Standard Error	Z	*p*
On Social Support				
Of Age	−0.015	0.058	−1.859	0.063
Of Education	−0.109	0.058	−1.400	0.161
Of Use of GS	0.181	0.061	2.971	0.003
On Geriatric Depression				
Of Age	0.226	0.054	4.157	<0.001
Of Education	−0.263	0.055	−4.803	<0.001
Of Use of GS	−0.063	0.059	−1.068	0.285
Of Social Support	−0.433	0.059	−7.328	<0.001
Direct effect	−0.063	0.059	−1.068	0.285
Indirect effect	−0.078	0.029	−2.687	0.007
Total effect	−0.142	0.060	−2.350	0.019

## Data Availability

The data presented in this study are available on request from the corresponding author. The data are not publicly available due to privacy regulations.

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
