# Peer review of "Perceived Social Support Mediates the Relationship between Use of Greenspace and Geriatric Depression: A Cross-Sectional Study in a Sample of South-Italian Older Adults"

_ijerph, 2023, doi:10.3390/ijerph20085540_

Round 1

Reviewer 1 Report (Previous Reviewer 2)

I thank the authors for the vast modifications of their manuscript.

The reanalysis of the data makes more sense to the overall topic. However, there are several typos, punctuation mistakes, and capital letter use in the text that needs to be corrected. 

Table 2 can be deleted as all the information is given in the text anyway.

The titel should indicate the country where the study was conducted. Also, the findings are closely connected to the population studied. So, a major weakness of the manscript is the lack of generalizability and representativeness. This aspect should be discussed in more depth. The limitations sections shows us that the study left out a lot of vital variables to offer a comprehensive approach to a delicate topic. Also, the public health relevance of this paper is not clearly outlined, in view of the name of the journal where the work was submitted. So, theoretical and practical implications of the results are lacking. E.g. if I would be a policy maker, I would not know what to do with the findings of this study, especially as the concepts of mental health, depression, depressive symptoms and geriatric depression are hard to capture for non-health professionals.

Author Response

Q1: Thank the authors for the vast modifications of their manuscript. The reanalysis of the data makes more sense to the overall topic. However, there are several typos, punctuation mistakes, and capital letter use in the text that needs to be corrected. 

R1: Thank you for your appreciation. We agree with you that the data reanalysis has improved the manuscript. We would like to thank you for noticing us about typos, punctuation mistakes and capital letters. We have revised the manuscript in order to correct them.

Q2: Table 2 can be deleted as all the information is given in the text anyway. 

R2: Thank you. We removed it.

Q3: The title should indicate the country where the study was conducted.

R3: Thank you for your suggestion. We modified it as follows: “Perceived social support mediates the relationship between use of greenspace and geriatric depression: a cross-sectional study in a sample of South-Italian older adults.”

Q4: Also, the findings are closely connected to the population studied. So, a major weakness of the manuscript is the lack of generalizability and representativeness. This aspect should be discussed in more depth.

R4: Thank you for pointing out this issue. We have expanded the discussion taking into account the lack of generalizability and representativeness of our results as follows, lines 292-295: “Moreover, since study results refer to a specific population study (i.e., older adults from the South of Italy) and to a relatively small sample size, these findings may suffer from a lack in generalizability to other population target (e.g., young and mature adults)” and lines 309-312: “This issue should be addressed in future studies. This study may be replicated: a) exploring the proposed structural relationship in different population target (e.g., young and mature adults)”.

Q5: The limitations sections show us that the study left out a lot of vital variables to offer a comprehensive approach to a delicate topic. Also, the public health relevance of this paper is not clearly outlined, in view of the name of the journal where the work was submitted. So, theoretical, and practical implications of the results are lacking. E.g., if I would be a policy maker, I would not know what to do with the findings of this study, especially as the concepts of mental health, depression, depressive symptoms, and geriatric depression are hard to capture for non-health professionals.

R5: Thank you for highlighting these issues. As you suggested, we have improved theoretical and practical implications of our study as follow, lines 332-341: “Furthermore, these results may be of interest of policy makers in light of the relevance of geriatric depression for the public expenditure. Indeed, geriatric depression is recognized as a significant public health issue which is strictly associated with economic burden for society. Adequate strategies for prevention of geriatric depression are needed. For instance, in order promote mental health and healthy lifestyle during the ageing, age-friendly environments and tailored nature-based solutions could be a relevant target to be pursued by policy makers. Moreover, social policies could consider the beneficial role of greenspace on mental health and these findings could be addressed in future urban planning.”.

Reviewer 2 Report (Previous Reviewer 1)

Greetings to the authors

The text has been improved by the addenda introduced by the authors, mainly to clarify that in subsequent investigations keys that have not been used at this time will be used. Even so, its impact is mitigated by what we commented in the previous review: the lack of solidity of the variables selected to address the object of study and the need to introduce a qualitative perspective to face the challenges of the analyzed phenomenon.

Having said this, the work is neat and, from our point of view, suitable for publication.

Congratulations. 

Author Response

Q1: The text has been improved by the addenda introduced by the authors, mainly to clarify that in subsequent investigations keys that have not been used at this time will be used. Even so, its impact is mitigated by what we commented in the previous review: the lack of solidity of the variables selected to address the object of study and the need to introduce a qualitative perspective to face the challenges of the analyzed phenomenon.

R1: Thank you for introducing this issue. We agree with you that a qualitative perspective may allow us an improved analysis of the phenomenon. Unfortunately, we are not able to deduce qualitative variables from our measures in the present study. Nevertheless, we are going to address a qualitative perspective in future studies, as you suggested.

Q2: Having said this, the work is neat and, from our point of view, suitable for publication. Congratulations.

R1: Thank you very much for you appreciation.

Reviewer 3 Report (New Reviewer)

Dear Authors,

I congratulate you for the effort you have put into the writing of the manuscript. In the attached document you can see the comments of my review.

Kind regards

Author Response

Q1: The first-person plural is used to describe information. In line 116 it appears: "we used a single item...". In other lines the possessive determinant "our" is also used. In scientific texts the third person singular/plural or indirect style should be used. The entire manuscript should be revised in this respect since this error is maintained in all sections.

R1: Thank you. We have modified the whole manuscript following your indication.

Q2: It is necessary to change the way of citing in the manuscript, eliminating the date of some references. For example, in line 54 it says: "Banay et al. (2018) [25]...". It would be correct to read: "Banay et al. [25]...".The entire manuscript should be revised in this respect, since this error is maintained in all sections.

R2: Thank you. We have modified the whole manuscript with respect to the way of citing references.

Q3: The abstract should include some quantitative results to attract readers to read the whole manuscript. Values such as: CFI, RMSEA... can be included.

R3: Thank you for your suggestion. We added results and fit indices in the abstract.

Q4: This section is very well elaborated, congratulations!

R4: Thank you very much for you appreciation.

Q5: In line 108 appears: "ds". Check whether to write "ds" or "sd".

R5: Thank you. We have modified it and we went for “sd”.

Q6: In section 2.2.2. it is mentioned that the PSSQ items were translated. The translation process should be explained, for example: use of a back translation model between translators with official qualifications.

R6: Thank you. We have better specified the translation process as follows, lines 131-132: “The items included in the original English validation were translated in Italian using the forward back translation method with the support of translators with qualifications.”

Q7: On lines 140-141 it is mentioned that the GDS instrument is also validated in the Italian population. The reference in which the instrument was validated should be placed next to it.

R7: Thank you for your suggestion. We modified it.

Q8: A figure with the proposed theoretical model should be included in section 2.3. In the following manuscript you can see an example of how to do it: https://www.mdpi.com/1660-4601/19/23/16177

R8: Thank you for your suggestion. We added it in section 2.3.

Q9: Section 2.3 statistical analysis and table 2 of the results section. The authors include the CFI, SRMR and RMSEA values. It would be convenient to include other indexes to test the goodness of fit of SEM, such as: normal fit index, incremental fit index and Tucler-Lewis index. An example of how to do this can be seen in the following manuscript: https://www.mdpi.com/1660-4601/19/23/16177.

R9: Thank you for noticing it. We have included among indexes useful to test the goodness of fit of SEM the Normed Fit index, the Incremental Fit index and Tucker-Lewis index.

Q10: Line 161 states that the CFI value should be ≥0.9 to consider an acceptable goodness of fit. The authors should review the information in reference number 45 since other publications consider that this value should be ≥0.95.

R10: Thank you for pointing out this issue. We agree with you in considering a CFI ≥0.95 as the cut-off value for a good fit. The Comparative Fit index (CFI) in our model is 0.934. As suggested by Hu & Bentler (1999) a CFI ≥0.90 is recognized as the conventional rule of thumb for this index and it has been considered and shared for a long time in the scientific community as an acceptable fit value. Further studies have recognized a CFI value ≥0.95 as indicative of good fit. The same goes for other common fit indices (e.g., TLI, NFI). There is an open debate about a) the usefulness of fit indices for assessing structural equation models and b) the employment of strict vs lenient cut-off values. Indeed, the adoption of strict values ensure that the model explains the maximum possible amount of variability in the data, thus capitalizing on that specific dataset with the risk of 1) poor generalizability to other dataset and 2) eventual overfitting, while the adoption of lenient values explains less amount of variability but reduces the risk of committing type II error, i.e. rejecting models that would be “good” but for some contingent reasons does not fully reach all the cut-off values for being considered as good models. We are aware of these issues, and we modified the text both in the subsection “2.3. Statistical Analyses” and in the subsection “3.2. Model testing” in order to be more cautious in reporting the results.

Q11: The note to figure 1 includes: p < 0.05. This information should be eliminated because this value is not considered in the figure.

R11: Thank you. We have removed it.

Q12: The meaning of TLI is included in the note to Table 2 but that abbreviation does not appear in the table.

R12: Thank you. We have removed the whole table 2, as suggested by the first reviewer.

Q13:  In Table 3 there are several p-values of 0.000. These values do not exist statistically. No matter how small they are, they can never be 0.000. The value used should be modified to ≤ 0.001. This comment should be taken into account to modify the information written in lines 214-223.

R13: Thank you. We have modified it.

Q14: The first paragraph mentions the aims of the research, the following three paragraphs contain a brief discussion and, subsequently, the limitations and future lines of research are mentioned. The section devoted to the discussion of the results of the study with those of the scientific literature is very brief and should be longer. Here is an example of how this extension could be extended. In the reference I discussed in comments 8 and 9 of this review paper, one can see how adolescents' perceived social support may be a mediating factor between physical activity and screen time. This would have some similarity with the results of the manuscript revised for publication.

R14: Thank you for pointing out this issue. In light of your suggestion, we have expanded our discussion as follows: lines 249-281: “Previous studies revealed that visiting and spending time in greenspace provided improvements in perceived mental health [50–52]. For instance, van der Berg et al. found positive association between spending time in greenspace and mental health and vitality. This study confirmed that the aforementioned association was independent of cultural and climatic contexts, as well. Furthermore, White et al. [52] found that the likelihood of reporting wellbeing was greater in people who spent at least 120 minutes per week in greenspace than in those who reported less time spent, even if positive association peaked for people who spent 300 minutes per week in greenspace. As opposed to, in the present study it was observed that use of greenspace was not directly associated with geriatric depression symptoms. Direct contact with urban greenspace may be not a requisite for improving health. Indeed, it was observed an indirect association between use of greenspace and geriatric depression. Our findings confirm the general hypothesis that contact with urban greenspace provided mental health benefits through several underlying mechanisms. This is in line with previous results on indirect association between use of urban greenspace and mental health [53] which indicated that visiting urban greenspace was associated with greater human wellbeing through several intervening variable (e.g., ego-depletion). Specifically, it was observed the mediating role of perceived social support in the association between contact with urban greenspace and geriatric depression as supported by previous research [35,54]. Dadvand et al. [54] found that perceived social support mediated the association between greenspace exposure and general health. Social support and perceived social support were found as mediators in the relationship between greenspace and mental health in several studies as confirmed by a recent review [35]. One possible explanation for the underlying mechanism of perceived social support regards the well-established social benefits (i.e., social interactions, social cohesion) of contact with urban greenspace. One may suppose that social interaction opportunities offered by the use of greenspace play a crucial role in promoting neighborhood social cohesion [55], thus enhancing perceived social support of older people”.

Q15: This section is very well elaborated, congratulations!

R15: Thank you very much for you appreciation.

Round 2

Reviewer 1 Report (Previous Reviewer 2)

thanks for modifying your manuscript!

Reviewer 3 Report (New Reviewer)

Dear authors,

I would like to congratulate you on your manuscript. The scientific quality of the manuscript has improved considerably after making the corrections and improvements indicated by the reviewers.

I consider that no further modifications are necessary. 

Kind regards

This manuscript is a resubmission of an earlier submission. The following is a list of the peer review reports and author responses from that submission.

Round 1

Reviewer 1 Report

The article analyzes the impact of green spaces on geriatric depression in the metropolitan area of Bari (Italy). The subject is really interesting, especially when the analysis is crossed with other social variables such as perceived social support. Congratulations to the authors.

In spite of the clear scientific interest that we can find behind this topic, the execution of the article presents a deficit that we consider should be corrected so that it becomes a significant contribution to the field of psychology. From our point of view, measuring the impact of green spaces on geriatric depression through the variables 'frequency of visit to these spaces' and 'time spent in them' does not provide us with an exhaustive look at what effect green spaces can have on geriatric depression. We recommend that they use some qualitative variable, because this is the way in which awareness of this impact could be gained.

So that they do not have to do a total reconstruction of the work, we recommend that the authors develop a series of indicators linked to the variables indicated so that the impact of green spaces on people suffering from depression in geriatric age can be effectively assessed.

Best regards,

Author Response

Q1: The article analyzes the impact of green spaces on geriatric depression in the metropolitan area of Bari (Italy). The subject is really interesting, especially when the analysis is crossed with other social variables such as perceived social support. Congratulations to the authors.

R1: Thank you for your words of appreciation for the manuscript.

Q2: In spite of the clear scientific interest that we can find behind this topic, the execution of the article presents a deficit that we consider should be corrected so that it becomes a significant contribution to the field of psychology. From our point of view, measuring the impact of green spaces on geriatric depression through the variables 'frequency of visit to these spaces' and 'time spent in them' does not provide us with an exhaustive look at what effect green spaces can have on geriatric depression. We recommend that they use some qualitative variable, because this is the way in which awareness of this impact could be gained. So that they do not have to do a total reconstruction of the work, we recommend that the authors develop a series of indicators linked to the variables indicated so that the impact of green spaces on people suffering from depression in geriatric age can be effectively assessed.

R2: Thank you for your suggestion. We agree with you on this point. The present study proposes an initial modeling which does not claim to be definitive and exhaustive on the topic but could serve as a basis for subsequent studies. Surely, qualitative variables could enrich the comprehension of the relationships between quantitative measures of greenspace and mental health outcomes. Although we are not able to deduce qualitative variables from our measures which were only based on standardized tools, we can speculate that a higher use of greenspace and a higher time spent in greenspace may correspond to a higher pleasantness of greenspace. Therefore, we addressed this point in limitations and future directions section as follows (lines 285-301): “Lastly, in our study we left out greenspace attributes (e.g., pleasantness, aesthetic quality, layout) as well as individual characteristics (e.g., connectedness to nature, urban-nature orientedness) which may be relevant predictors of propensity to visit urban greenspace and of mental health benefits of greenspace [58–61]. For instance, as suggested by previous studies [58,61] visit frequency in urban greenspace and mental health benefit of greenspace may be influenced by park features, such as size, layout, pleasantness, and items involved in the park area. Furthermore, Ojala et al. (2019) [60] suggested that individual characteristics (e.g., urban-nature orientedness) may affect restorativeness provided by contact with greenspace. This issue should be addressed in future studies. This study may be replicated: a) using a Global Positioning System (GPS) device to objectively measure the time that participants spend in greenspace, b) using objective data for assessing geriatric depression symptoms (e.g., medical prescriptions, psychiatric diagnosis) and for social support (e.g., sociometric tools), and c) considering some individual characteristics, such as measures of spatial cognition based on familiar [63] and unfamiliar locations, as well as d) greenspace attributes.”

Reviewer 2 Report

This manuscript is on the impact of perceived social support on use of greenspace and geriatric depression in a sample of Apulian mature and older adults. Although the tooic is of certain relevance, the article as several flaws that limit its merit to readers. 

I suggest to shorten the title, also accounting for the fact that the difference between mature adults and older people as well as elderly was not explained in the text, and especially as also middle-aged people were participants. Unfortunately, this is a severe shortcoming of the method, significantly reducing the representability and generalizability of the findings.

The introduction would profit from a detailed conceptual framework.

In the abstract, the finding that the use of greenspace was indirectly associated with geriatric depression needs explanation. What is the nature of this indirect pathway?

Could you explain why the depression rate is higher in in Europe compared to the world?

The rationale behind the study is not sufficiently explained, e.g. in respect to the study design and the country of origin of the study.  

The authors state that evidence is lacking in studies on the role of human engagement with greenspace. This is contradicting the aforementioned existence of a vast amount of studies in this field.

There are many style errors and typos e.g. line 94 and 206, among others. For instance, “To our knowledge, limited evidence on older adults is available” (sic).

The format of the tables reduce their comprehensability.

The recruitment process is unclear. What is the role of proxy informants, their recruitment etc.

When was the study conducted?

The questionnaire in Italian was neither pretested nor validated. Also, the questionnaire items should be reported in more details to allow for replication of the study.

Socio-demographic variables as well as descriptive results of the survey are missing, which do not allow for assessing the validity of the paper. Practical and theoretical implications for other settings and countries are lacking.

In sum, the article has many flaws throughout the text that limit representability and generalizability of the findings.

Author Response

Q1: This manuscript is on the impact of perceived social support on use of greenspace and geriatric depression in a sample of Apulian mature and older adults. Although the topic is of certain relevance, the article as several flaws that limit its merit to readers. 

R1: Thank you for this general comment. We tried our best to fix all the issues raised and we hope that now the manuscript has improved its content.

Q2: I suggest to shorten the title, also accounting for the fact that the difference between mature adults and older people as well as elderly was not explained in the text, and especially as also middle-aged people were participants. Unfortunately, this is a severe shortcoming of the method, significantly reducing the representability and generalizability of the findings.

R2: Thank you very much. We shortened the title as you suggested. We better specified the difference between mature and older adults in the text of the manuscript, and we have been consistent with the terms used for describing the sample. We also added a specification on the study design in the title. The new title is as follow: “Perceived social support mediates the relationship between use of greenspace and geriatric depression: a cross-sectional study”.

Q3: The introduction would profit from a detailed conceptual framework.

R3: Thank you for your suggestion. Our conceptual framework is driven by the two main theories in environmental psychology regarding the association between nature exposure and human health, namely Attention Restoration Theory (ART) and Stress Reduction Theory (SRT). We have provided a more detailed conceptual framework in the introduction section. We better stressed the theory related to our study as follows (lines 64-75): “Overall, Markevychh et al. [27] summarized in a unifying conceptual framework several mechanisms underlying mental health benefits associated to greenspace. One of the most important concerns the restoring capacities of greenspace, i.e., the so-called “restorativeness”, based on the Attention Restoration Theory (ART) and on the Stress Reduction Theory (SRT) [28,29]. The ART [29] provides a theoretical framework as well as related predictions for the cognitive impact of urban greenspace. Similarly, the SRT [28] provides theory and predictions for the affective impact of urban greenspace. Specifically, the SRT assume that contact with urban greenspace through the activation of parasympathetic system reduce stress, autonomic arousal, and enhance positive emotions, since we have evolved with an innate preference for green environments [28,30].”

Q4: In the abstract, the finding that the use of greenspace was indirectly associated with geriatric depression needs explanation. What is the nature of this indirect pathway?

R4: Thank you. Following your suggestion, we rephrased as follow (lines 24-25): “Results showed that the use of greenspace was inversely associated with geriatric depression through perceived social support.” Indeed, the coefficient of the indirect pathway was negative.

Q5: Could you explain why the depression rate is higher in in Europe compared to the world?

R5: Thank you for this question. As suggested by Hu et al. (2021) the rate of depression in older people was estimated to be 28.4% in the World and 21.1% in Europe. The lower rate of depression in Europe with respect to that of the World could be explained by its higher economic development and widespread supportive policies for mental health (Hu et al., 2021).

Q6: The rationale behind the study is not sufficiently explained, e.g. in respect to the study design and the country of origin of the study.  

R6: Thank you for your suggestion. As you have indicated, we have provided a better explanation of our rationale in the introduction section. See Q3 and lines 89-97 “In light of the above mentioned lack in previous research a cross-sectional study was carried out. The present paper attempts to extend knowledge on the relationship between self-reported direct contact with greenspace (i.e., use of greenspace as indicator) with self-reported depression symptoms and exploring one of the potential underlying mechanisms of this association (i.e., perceived social support). The aims of this study were: a) to investigate the relationship between use of greenspace and depression symptoms during mature adulthood and ageing, and b) to inves-tigate the role of perceived social support in the aforementioned relation-ship.”

Q7: The authors state that evidence is lacking in studies on the role of human engagement with greenspace. This is contradicting the aforementioned existence of a vast amount of studies in this field.

R7: Thank you for noticing it. At the beginning of the introduction (lines 49-55) we stated that there were a growing number of studies on the association between contact with greenspace and several health outcomes. Subsequently, we highlighted few evidence and few studies on the association between contact with greenspace and mental health and depression symptoms respectively. Actually, at lines 77-78 we argued that few evidence on the role of human engagement with greenspace were available and that sentence may have generated a contradiction with the previous ones. We deleted such sentence and now the paragraph is as follow: “Besides, human engagement with greenspace includes indirect contact with greenspace, such as viewing greenspace from a distant point, and direct contact with urban greenspace (i.e., use of greenspace), such as visiting and spending time in greenspace [36]. Nevertheless, evidence on the effect of visiting greenspace on mental health were still limited [37].”

Q8: There are many style errors and typos e.g. line 94 and 206, among others. For instance, “To our knowledge, limited evidence on older adults is available” (sic).

R8: Thank you very much. We tried to fix several grammar issues and typos throughout the whole manuscript.

Q9: The format of the tables reduce their comprehensability.

R9: Thank you for noticing it. Table formatting was modified by the journal submission system. We try to provide in this version a correctly formatted table.

Q10: The recruitment process is unclear. What is the role of proxy informants, their recruitment etc.

R10: Thank you for this comment. We better clarified the recruitment process of our study as follows (lines 100-105) “A convenience sample of mature and older adults’ resident in the metropolitan area of Bari (Apulia) were recruited between January 2022 and April 2022, with the support of a proxy informant. The latter were recruited among undergraduate students, postgraduate students, and trainees. They were requested to contact and invite mature and older adults to be enrolled in the study.”

Q11: When was the study conducted?

R11: Thank you for this question. The study was conducted in the early months of 2022 (from January 2022 to April 2022). We added this information in the text as follow (lines 100-102): “A convenience sample of mature and older adults’ resident in the metropolitan area of Bari (Apulia) were recruited between January 2022 and April 2022.”

Q12: The questionnaire in Italian was neither pretested nor validated. Also, the questionnaire items should be reported in more details to allow for replication of the study.

R12:  Thank you for your suggestion. As you highlighted, the Duke-UNC functional social support questionnaire (FSSQ) was not validated in Italian population. Nevertheless, in our data the Cronbach’s a and the McDonald’s w of FSSQ were indicative of a good reliability (respectively 0.85 and 0.86). Moreover, the factors loadings of the total score for each subscale on the latent variable (i.e., perceived social support) were good. Following your suggestion, we reported more details about the Duke-UNC functional social support questionnaire (FSSQ) in supplementary materials.

Q13: Socio-demographic variables as well as descriptive results of the survey are missing, which do not allow for assessing the validity of the paper.

R13: Thank you for this comment. Socio-demographic variables had been previously added in Table 1, which was wrongly formatted.

Q14: Practical and theoretical implications for other settings and countries are lacking.

R14: Further, we added practical and theoretical implication for other settings and countries in conclusion section, lines 311-314: “Future research may take into account differences in levels of urbanization and in rates of urban greenspace in other Countries to confirm and extend the present results”.

Round 2

Reviewer 2 Report

I thank the authors for modifying the manuscript. However, unfortunately, the most severe errors in the method are still present, as an age group including people younger than 60 years of age is not suitable to test for the study question including onset of geriatric depression using a cross-sectional design. Again, there is a constant mixing up of terms such as mature age, older age etc. in the text. I suggest removing the younger age group from the analysis or adapting the study design. Otherwise, the interpretation of the statistical procedures do not make sense.

Also, the format of the mansucript should be checked to align with the journals requirements.

Author Response

I thank the authors for modifying the manuscript.

R: Thank you very much for your previous and current enlightening comments and suggestions. We tried to address each point exhaustively. Please find the modifications with respect to the previous version green-highlighted.

However, unfortunately, the most severe errors in the method are still present, as an age group including people younger than 60 years of age is not suitable to test for the study question including onset of geriatric depression using a cross-sectional design.

R: Thank you for this comment. We have thought over this point and we agree with you, our initial idea to collapse all age groups within a single model was probably too tentative and theoretically not justified, since we use some psychological tools which were appropriated only for elderly people. Thus, we revised the method section and in particular the specific parts related to sample composition as well as the results section with respect to the changes in model coefficients. Tables and figure were revised as well.

Again, there is a constant mixing up of terms such as mature age, older age etc. in the text.

R: Thank you very much. We tried to be more consistent with respect to previous version of the manuscript. We decided to use only the expression “older adults” for designating participants included in the study.

I suggest removing the younger age group from the analysis or adapting the study design. Otherwise, the interpretation of the statistical procedures do not make sense.

R: Thank you very much. This was the focal point of it all. We removed the younger age group from the sample and, subsequently, from the statistical analysis. Then we rerun the SEM including only participants with 60 years of age or more. The general pattern of the results did not vary. We modified the text, the tables and the figure accordingly.

Also, the format of the manuscript should be checked to align with the journals requirements.

R: Thank you. We checked the manuscript in order to align it with the journal requirements.